# Impact of Burnout on Daily Activities from an Occupational Therapy Perspective: A Serial Mediation Model with the IDA Scale

**DOI:** 10.3390/bs12110426

**Published:** 2022-10-31

**Authors:** Alicia Pérez-Santiago, Luis-Javier Márquez-Álvarez, José Antonio Llosa, Estíbaliz Jiménez Arberas

**Affiliations:** Faculty Padre Ossó, University of Oviedo, 33008 Oviedo, Spain

**Keywords:** occupational therapy, activities of daily living, burnout syndrome, mental health

## Abstract

Background: Burnout syndrome is one of the most frequent health complications among workers. Acknowledging the work perspective as something basic and essential in a person’s life means that this disorder can have huge implications in their most basic daily activities. Methods: A cross-sectional, quantitative observational design was conducted with data from Spanish workers. A serial mediation model was applied to study the relationship between daily activities and burnout syndrome. For this purpose, the IDA scale was developed. Conclusions: The results show us that peoples’ work situation has an impact on their daily life. There is quantitative evidence of the impact on daily life occupations and how it further decreases the levels of health and well-being of the person, on their independence and, consequently, on their quality of life.

## 1. Introduction

Burnout syndrome (BS) is one of the most prominent work-related health threats. It was defined by Perlman and Hartman [1] as the response to chronic emotional stress characterised by three components: emotional or physical exhaustion, lowered job productivity and depersonalisation or a lack of concern for others. For Maslach, burnout emerges as a response at the individual level, i.e., BS is an internal psychological experience involving feelings, attitudes, motives and expectations. This situation creates the emergence of symptoms that impact emotional exhaustion (EE), depersonalisation (DP) and personal accomplishments (PA) [2,3]. According to Golembiewski et al. [4], DP is a trigger for the other two. In the construct of burnout, based on the classic theories of work-related stress [5], DP is also the characteristic variable for the definition of the phenomenon [6].

Even if occupational health services are entrusted with essentially preventive functions and are responsible for advising the employer, the workers and their representatives in undertakings on the requirements for establishing and maintaining a safe and healthy working environment [7], on many occasions, this protection is not carried out, leaving employees unprotected and at the mercy of factors that pose a risk that is not only physical but also emotional.

Data indicate that its prevalence is 20% in the active working population [8], although these data vary in different sectors of the labour market and in different countries [9,10], associating it with 55% of sick leave and with economic costs of more than EUR 20 billion in European countries [11].

There are two different perspectives from which this syndrome is viewed: a clinical perspective and a psychosocial perspective [12]. The clinical perspective assumes the subject and the job as two atomic elements that are practically independent, while the psychosocial perspective understands burnout as a phenomenon that emerges in the interaction of working conditions and the person him/herself. Some studies refer to the existence of several factors that may favour BS appearance. This is not only at the level of the job, but there are also social, environmental and personal factors that increase the risk of a person suffering from the most characteristic symptoms [13,14].

In occupational therapy, work is recognised as something basic and essential, not only because of the financial remuneration, which is necessary in people’s daily lives in order to have an optimal quality of life, but also because of its repercussions on a personal level [15]. Occupational therapists play a crucial role in the process of returning to work thanks to their ability to improve occupational performance, defined as the accomplishment of the selected occupation resulting from the dynamic transaction among the client, their context and the occupation, which also can include work [16]. In addition, they have the ability to holistically assess the physical, cognitive, emotional and social skills and abilities that are intertwined in such work [17].

Obtaining a job gives people increased self-esteem, identity formation, the development of physical, social and cognitive skills and personal and professional growth. All of these are key to the maintenance of health in the population [18].

Occupation in this discipline is understood as a key term that is necessary to understand the relationship between health and work. It is defined as the everyday activities that people engage in as individuals, in families and with communities to occupy time and bring meaning and purpose to life. Occupations include things that people need to, want to and are expected to do [19]. This concept encompasses all those activities that are performed on a daily basis and are carried out in specific contexts and environments. Work itself is typified in the Occupational Therapy Practice Framework 4th edition (OTPF-4) as including activities such as the recognition of activities of interest for a subsequent job search, job searches and acquisitions or job performance itself [20]. It is therefore very important to emphasise the huge impact that employment has on people’s lives. Being unemployed or having a job in which one works under bad conditions or where minimum requirements are not fulfilled can be very damaging to health.

Taking into account the impact it has on the daily lives of people with symptoms such as stress or anxiety and its possible impact on participation and engagement in the activities of daily living, the evidence still does not gather enough data on this aspect. Few articles directly link these two fields, mostly related to BS in caregivers [21,22]. There is still no clear evidence to extrapolate the impact of burnout on the person’s daily activities, understanding them as the daily life activities people find purposeful, meaningful and necessary for an independent living.

Therefore, the purpose of this study is twofold. The first objective is to generate a statistically valid assessment instrument that analyses the level of occupational performance among the population in active employment. The second objective, having such an instrument, is to study the relationship between phenomena related to the quality of work life, such as burnout, and people’s occupational performance. These two consecutive objectives in this research provide a valid tool for the relationship between working conditions and a person’s occupational performance. In other words, it provides a broad understanding of the way in which working conditions impact a person’s life and well-being in a holistic way.

## 2. Materials and Methods

### 2.1. Sampling and Procedure

The data for this study were collected from a sample of people in active employment in Spain. For this study, the use of data from retired people, unemployed people or people without a prior work record was not considered. The sample selection that participated in this project was chosen using the snowball sampling technique.

Table 1 shows the descriptive data of the n = 150 workers. The population studied is predominantly female (76.7%), and the predominant age range is 20–25 years (28.7%). The length of service is highly distributed among those who have been working for 5 years or less (58.6%) and within the health and social care sector (52.7%).

The survey with the measurements was launched through different social networks and through direct contact with people who met the inclusion criteria. It was distributed by providing a QR code and a link for its completion. The data collection period ran from December 2021 to April 2022, with a total of 150 participants.

### 2.2. Measurements

For the purpose of the study, three categories of data were collected: (a) socio-demographic data on age, sex, work sector and length of service; (b) data on BS, using the Maslach Burnout Inventory (MBI) [23] in its Spanish version [12,24]; (c) data on daily activities, using the Impact on Daily Activities (IDA) Scale, created for this purpose.

The MBI has 22 items divided into three sub-scales: emotional exhaustion (EE), depersonalisation (DP) and personal accomplishment (PA). It includes statements that are assessed with a Likert Scale of 0 to 6 values related to the frequency with which the phenomena described appear, with 0 being never and 6 being every day. The diagnosis of BS implies values higher than 26 in the EE subscale, higher than 9 in the DP sub-scale or lower than 34 in PA. The MBI has been studied on numerous occasions, with endorsements of reliability and validity. These assessments of the scale are made in a trifactorial manner, weighting the three sub-scales separately and jointly.

For this study, the IDA Scale was developed. This questionnaire was formulated as a series of statements to measure the level of occupational performance in activities of daily life in the population of active workers. This tool was created based on the OTPF-4 occupational inventory [20], assessing each statement (Appendix A) using a Likert scale from 1 to 5 depending on the level of agreement (1 = strongly disagree; 5 = strongly agree).

### 2.3. Data Analysis

Descriptive and correlational analyses were performed using the programs IBM SPSS v.25 (IBM, Madrid, Spain) [25] and JASPv.0.16.3 (JASP Team) [26]. Sociodemographic variables were taken as categorical variables, while MBI and IDA were taken as continuous variables. Descriptive analyses were carried out to obtain population frequencies, proportions, averages and standard deviations. Pearson’s coefficients were performed for the comparison of quantitative variables.

The validation process of the IDA was divided into three stages: (a) a pilot test and an initial measurement of reliability (α > 0.75) for use in the present study; (b) an exploratory factor analysis of polychoric correlations (EFA); (c) A confirmatory factor analysis (CFA), whose adequacy was estimated with the fit indicators CFI, TLI, GFI, SRMR and RMSEA. For this process, the results of the scale were analysed using the IBM SPSS v.25 [25] and JASP v.0.16.3 [26] software. The reliability of the scale was analysed with both Cronbach’s alpha and McDonald’s Omega. Finally, criterion validity was analysed through correlational analyses between the test being validated and the sub-scales of the MBI.

Serial mediation analysis was carried out using R v.4.2.1. (R Core Team, Vienna, Austria) [27]. This is a structural equation path analysis [28] in which a serial mediation model was proposed between aspects of burnout and the impact on daily life, where the impact of DP on daily activities is mediated by the constructs of PA and EE. This design is built based on burnout model from Golembiewski et al. [4].

## 3. Results

### 3.1. Validation of the IDA Scale

After dropping item 7 (I no longer enjoy taking care of my pet or I find it unthinkable to have one.) due to its low representativeness in the sample, an EFA [29,30,31] was performed on the 10 remaining items with the Factor software, using an optimal implementation of Parallel Analysis [32]. After an analysis of polychoric correlations with 500 boot samples and a Bootstrap confidence interval of 90%, it was decided to drop item 8 (I rest poorly or I have trouble falling asleep.) since the communality of item 8 was low (0.340) and its factorial weight (0.583) was substantially lower than that of the rest of the items.

The EFA was repeated with the remaining nine items, yielding a good fit to the analysis (KMO = 0.875; Bartlett’s statistic = 449.8 (df = 36; *p* < 0.00)). The UniCo index = 0.986 and the Mieral index = 0.287 indicate that the nine items analysed maintain a unidimensional structure. The unidimensionality criterion with both indicators is a value above 0.95 in the case of UniCo and below 0.3 in the case of Mireal. Moreover, all factor weights of the nine items are above 0.7 (Table 2).

The reliability of the scale was calculated using Cronbach’s alpha and McDonald’s omega indices, with good results in both cases [33]: McDonald’s ω = 0.870 and Cronbach’s α = 0.869. Final version of the scale can be seen in Appendix B.

For the CFA [29,30,31], the factor with nine items was extracted using the ULS extraction method. The analysis of the goodness-of-fit indices reaffirms unidimensionality as an adequate test construct (CFI = 0.993; TLI = 0.990; RMSEA = 0.042; SRMR = 0.099; GFI = 0.987) [34] (Table 3). The final scale consisted of the nine items mentioned.

### 3.2. Validity and Relationship with Other Variables

The results of the scale in relation to the population show a non-normal distribution (Kolgomorov–Smirnov < 0.05), with a mean = 20.97(SD = 7.43). The Kruskal–Wallis test shows no significant differences between sociodemographic factors, similar to the findings of the MBI.

Criterion validity is analysed using Spearman’s Rho, given the non-homoscedasticity of the sample. Table 4 shows the intercorrelations between the MBI variables (DP, EE and PA) with the IDA scale. All sections of the MBI show statistical significance with IDA. It was observed that the EE of the MBI scale had a strong relationship with the impact on daily life and that PA was also negatively related to IDA.

### 3.3. Serial Mediation Model Analyses

The serial mediating role of the Personal Accomplishment (PA) and Emotional Exhaustion (EE) variables in this relationship was analysed. To do so, a model was designed in R Studio, with 5000 bootstrap samples (95% NC) and ML (maximum likelihood) estimator in the selected sample.

In the model, the dependent variable is IDA (Y), the independent variable is depersonalisation (X) and the mediators are personal accomplishment (M1) and emotional exhaustion (M″) (Table 5; Figure 1).

The indirect effect of serial mediation (b = 0.047; *p* < 0.05) is significant, so we must assume that the relationship between depersonalisation and the impact of BS on daily activities has an effect through the sequential intervention of personal accomplishment and emotional exhaustion. However, the role of the emotional exhaustion mediator is the most relevant, as it accounts for 75.9% of the total indirect effect explored in the model.

It is also interesting to note that, although the total effect (c) of depersonalisation on daily activities is significant (b = 0.436, *p* < 0.000), when the direct effect (cp) of this relationship is calculated in the model by removing the variability due to mediators, it is not significant (b = 0.026; *p* > 0.05). In other words, in order to understand the relationship between depersonalisation and the impact of BS on daily activities, it is necessary to have a low score in personal accomplishment and, in particular, a high score in emotional exhaustion.

## 4. Discussion

In its first objective, this study aimed to generate a validated instrument for the analysis of occupational performance in the working population. With this aim in mind, the IDA scale was created, which has a psychometric validation that makes it suitable for analysing the population in the Spanish context.

Of these initially collected occupations, there were two aspects removed from IDA: sleeping and pet care. Although pet care was removed due to its low representativeness in the results collected, there are several studies on the positive impact of pets with respect to employment [35,36,37]. However, results obtained by Jensen et al. [35] corroborate that there is a significant association between facility dog presence and personal accomplishment (0.42 standard deviations higher for personal accomplishment). Given that, according to our mediation model, personal accomplishment is the variable that has the lowest impact on daily life, the result is congruent with its elimination from the scale.

Regarding sleep, there are different studies that have studied its role in different work sectors. For example, in the study by Wolkow et al. [38], firefighters screening positive for sleep disturbances were more likely to have high EE, DP, low PA and a high degree of burnout than those who did not screen positive. Similar results were observed in the Cheng & Cheng study [39], where the findings showed that workers engaging in night shifts experienced poor mental health and reported more sleeping problems as compared with day workers. In this sense, it is likely that the initial hypothesis in setting the item was not correct, i.e., that sleep is not impacted by BS but is a cause of it. Thus, its exclusion from IDA in the measurements could fit with the idea that the scale measures the impact, not the cause. This would be congruent with studies that support the hypothesis that sleep deprivation is associated with clinical burnout, since burnout is a syndrome encompassing feelings of physical, emotional and mental depletion, and it is considered a consequence of employment environment stressors, without adequate recovery [40,41,42].

Secondly, the research sought to find out whether BS has an impact on the daily life occupations of people in active employment. To our knowledge, this is the first study to address this issue from the perspective of occupational therapy by means of a structural equation model that presents a serial mediation of the emotional exhaustion and personal accomplishment variables in the predictive capacity of depersonalisation on occupational performance. Of all the variables studied, the relationship with emotional exhaustion seems to be the most relevant for the model, having a direct impact on the person’s life.

As for sociodemographic factors, there are a large number of studies that discuss those factors that generate a predisposition [43]. However, on numerous occasions, they do not agree and deny that there is a link to certain variables such as sex or age [44]. This statement would be congruent with our sample results, where no demographic group was more representative of the presence or absence of BS.

In the literature, most articles focus on demonstrating the severity of this syndrome in work sectors, especially those related to health and education [45], but few talk about the impact it can have on people’s daily lives. Those that focus on this aspect only refer to the appearance of anxiety, depression or other types of mental health pathologies or behavioural changes [46]. The results provided in this study show that people’s employment situation has an impact on their daily lives. Therefore, when faced with a bad situation at work, there are negative repercussions in occupations.

The proposed serial mediation model also allows for the replication of the stablished relations between the different dimensions of burnout. Starting with the analysis of Golembiewski et al. [4], the presented path analysis in our study also considers depersonalisation as the trigger of the rest of the dimensions of the construct. The model of Golembiewski et al. [4] established that a high DP also implied a high EE and a low PA. In the displayed results, this relation is maintained: path A1 of the analysis points to a significant relationship (coefficient −0.41), i.e., the increase in the variable DP supposes a decline in the PA. The A2 effect, also significant, shows a direct relationship between DP and EE (0.98). However, even though Golembiewski et al.’s model is considered a classic model, its logic and nuances are still being discussed in the academic literature. The proposed sequence for dependence was empirically proven by other researchers but not necessarily in other analysed work contexts [47]. As a contribution to this discussion, our path analysis supports the logic on the model from Golembiewski et al. in the general population.

Another hint that encourages a deeper reflection of this model is the fact that our design has found that the indirect effect of PA (M1) does not have a significant impact on the displayed results. However, when both PA and EE are included (M1-M2), the impact becomes significant. However, it seems remarkable that the direct effect of DP on daily activities (cp) is not statistically significant. All of this has at least two implications: first, a parallel mediation model including the two variables determined as mediators (EE and PA) does not result in a statistically relevant association; second, the direct effect (cp) is not significant but contributes to the total effect.

We are also certain that, in between DP and daily activities, there are other variables intervening. In fact, the difference between the direct and indirect effects showed that 75.9% of the variance determined by DP in daily activities depends on the mediating variables. All of this helps to support the idea of DP as a trigger variable of burnout [4,48] and the idea that variables configuring dimensionalities of the burnout do not have to be observed as being independent [47].

The main limitation of the study was the lack of previous references. This research is based on a gap in the state of the art regarding how occupational therapy can improve people’s quality of life and work. However, this field still seems to be largely unexplored—hence the scarcity of scales and the obligation to create one for this purpose. On the other hand, the sample size has limited the results to a large extent. We believe that, with a larger sample size, a normal distribution could have been achieved, as the current distribution resembles this normality to a large extent. This would further extend the range of statistics and their representativeness.

Finally, the mediation model provided did not take into account the different sociodemographic data because they were not significantly related to the variables of interest. However, given that there is no consensus on the role played by these variables in BS, we consider that this field needs to be explored further.

## 5. Conclusions

Burnout syndrome is a disorder that begins in the work environment but transcends it. This study has quantitatively evidenced the impact that it has on daily life occupations. This impact further reduces the levels of health and well-being of the person, their independence and, consequently, their quality of life.

This study has shown that there is a relationship between burnout syndrome and occupations. The results show that people’s work situation has an impact on their daily lives. If the situation at work is bad, there are negative repercussions on occupations. It is important to create policies aimed at preventing the emergence of mental health-related pathologies and not only those related to physical health. In the country where this study was carried out, there are several assessments that study the appearance of psychosocial problems in different companies, but this study shows that these assessments are scarce and are progressing very slowly. Therefore, there is still a lack of programmes aimed at the prevention and treatment of this type of problem, where occupational therapy can make an impact thanks to its holistic vision of work.

## Figures and Tables

**Figure 1 behavsci-12-00426-f001:**
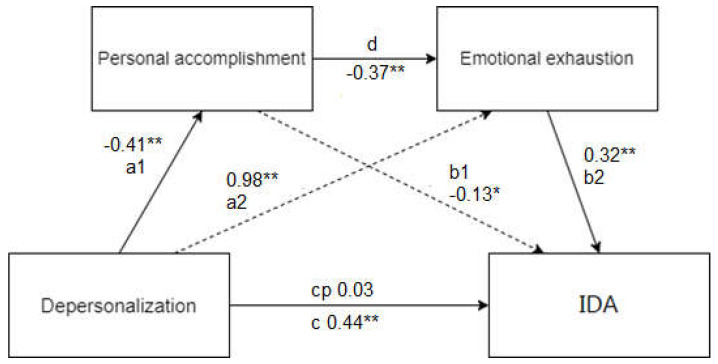
Serial mediation model of despersonalisation on daily activities through Personal Accomplishment and Emotional Exhaustion. * *p* < 0.05; ** *p* < 0.01.

**Table 1 behavsci-12-00426-t001:** Socio-demographic characteristics of the sample.

Variable		N (%)	Variable		N (%)
Sex	Women	115 (76.7)	Employment sector	Health and social care	79 (52.7)
	Men	34 (22.7)		Education	18 (12.0)
Age (years)	20–25	43 (28.7)		Hospitality and trade	17 (11.3)
	26–30	27 (18.0)		Leisure, events, culture	11 (7.3)
	31–35	12 (8.0)		Public admin.	3 (2.0)
	36–40	8 (5.3)		CIT and telecommunications	2 (1.3)
	41–45	10 (6.7)		Mining, construction, maintenance and automotive	8 (5.3)
	46–50	13 (8.7)			
	51–55	20 (13.3)		Housewife	1 (0.7)
	56 or more	14 (9.3)		Research	2 (1.3)
Length of service	1 year or less	41 (27.3)			
	1–5 years	47 (31.3)			
	6–10 years	17 (11.3)			
	11–15 years	9 (6.0)			
	16 years and over	33 (22.0)			

**Table 2 behavsci-12-00426-t002:** Factor loadings and reliability statistics.

Item	Factor Loadings	Reliability if Items 8 and 9 are Dropped		Item–Rest Correlation
		McDonald’s ω	Cronbach’s α	
1	0.772	0.859	0.860	0.548
2	0.732	0.860	0.857	0.576
3	0.782	0.851	0.850	0.652
4	0.716	0.857	0.856	0.587
5	0.712	0.861	0.858	0.559
6	0.745	0.856	0.855	0.604
7	0.772	0.857	0.855	0.592
10	0.789	0.847	0.847	0.683
11	0.732	0.854	0.852	0.624

**Table 3 behavsci-12-00426-t003:** CFA results.

Model	Χ²	df	Goodness-of-Fit Criteria Values
Baseline model	474,819	36	
Factor model	30,290	27	
**Index**	**Value**		
Comparative Fit Index (CFI)	0.993		≥0.95
Tucker–Lewis Index (TLI)	0.990		>0.95
Root mean square error of approximation (RMSEA)	0.042		≤0.10
Standardised root mean square residual (SRMR)	0.099		≤0.08
Goodness-of-fit index (GFI)	0.987		≥0.95

**Table 4 behavsci-12-00426-t004:** Spearman’s correlations among study variables; ** *p* < 0.01.

	MBI EE	MBI DP	MBI PA	IDA
MBI EE	1			
MBI DP	0.333 **	1		
MBI PA	−0.315 **	−0.215 **	1	
IDA	0.440 **	0.242 **	−0.276 **	1

**Table 5 behavsci-12-00426-t005:** Indirect effects of BS on daily activities.

		Indirect Effect	Boot SE	*p*	Bootstrapping 95% CI
Total Indirect Effects		0.411	0.081	0.000	[0.270; 0.589]
Ind effect M1: Despersonalisation → Personal Accomplishment → IDA	a1b1	0.052	0.034	0.12	[0.005; 0.141]
Ind Effect M2: Despersonalisation → Emotional Exhaustion → IDA	a2b2	0.312	0.069	0.000	[0.005; 0.141]
Ind Effect M1-M2: Despersonalisation → Personal Accomplishment → Emotional Exhaustion → IDA	a1db2	0.047	0.022	0.03	[0.015; 0.105]

## Data Availability

Not applicable.

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
