# Peer review of "Impact of Burnout on Daily Activities from an Occupational Therapy Perspective: A Serial Mediation Model with the IDA Scale"

_behavsci, 2022, doi:10.3390/bs12110426_

Round 1
Reviewer 1 Report
the study was well-done and well-written.
a minor suggestion - change the name of the OC scale and not use OC9. it is confusing in the tables if OC9 refers to the scale or to the OC9 item from table 2. A minor suggestion but clarity for the reader will improve.
Author Response
the study was well-done and well-written.
a minor suggestion - change the name of the OC scale and not use OC9. it is confusing in the tables if OC9 refers to the scale or to the OC9 item from table 2. A minor suggestion but clarity for the reader will improve.
Response: We thank the reviewer the compliments about our article. We appreciate the suggestion on renaming the scale, so we modified to “IDA” (for “Impact on daily activities”). Also, we added it in the title for better clarifications, now it reads: “Impact of burnout on daily activities from an occupational therapy perspective: a serial mediation model with the IDA scale”.
Reviewer 2 Report
Comments
1. The title of the manuscript is possibly misleading to readers. In the US, there is an occupation called “occupational therapy” – and your manuscript could mislead the reader into thinking it is about how occupational therapists would approach this issue of burnout.
2. You should explain “daily occupations” and “occupational performance” as early as you can in your manuscript. These terms (as used in your manuscript) are probably not familiar to many readers. Most readers probably think occupational performance means workplace performance rather than the homelife reflection. The thrust of your article only became clear when I saw the statements from the OC9 scale in Appendix A. I think you need to make it known upfront that the thrust of this manuscript is understanding the impact of working conditions (as measured through the components of burnout) on a person’s daily life and well-being.
3. Throughout the article, you inconsistently use the abbreviation “DP.” Most times you present it incorrectly as “PD”.
4. Line 38. Does “its” refer to BS? If so, simply use the term BS instead.
5. Table 2. On row 2 of the Table, I would restate it as: “Reliability if Items 7 and 8 are dropped.”
6. Major comment. In “structural equation modeling” studies like the one performed, when the components are assembled in a manner that is statistically relevant, there needs to be some explanation (or guesses) as to why this model works. For instance, the questions I have are:
a. Why is it serial mediation and not parallel mediation?
b. Why is depersonalization the independent variable of interest and personal accomplishment and emotional exhaustion the (serial) mediators? An argument could be made that emotional exhaustion should influence depersonalization and personal accomplishment.
What I am attempting to say is your research has provided a model without any reflection on burnout (in the discussion section) - the authors do not provide any explanation as to why one would expect the components to operate in a manner that the model indicates. At least a few paragraphs should be provided trying to explain the connections. Thus, you need to attempt to explain somehow the predictive capacity of depersonalization through the mediating variables of emotional exhaustion and personal accomplishment on occupational performance. Otherwise, the results are merely just results, and perhaps a better paper would have been to simply show the correlation of a burnout index (or individual components of burnout) with occupational performance.
Author Response
- The title of the manuscript is possibly misleading to readers. In the US, there is an occupation called “occupational therapy” – and your manuscript could mislead the reader into thinking it is about how occupational therapists would approach this issue of burnout.
R1. First of all, we would like to thank the reviewer the time in reading our manuscript. About the first comment, the actual purpose of this research is to stablish an approach to how burnout can impact in activities of daily living, as the main field of study in occupational therapy. In fact, three researchers of the team are occupational therapists with interest in this field. We think that the “occupation” and “occupational” terms can create confusion among the readers, because of this, we modified the term “occupations” by “activities of daily living” in most of the document.
- You should explain “daily occupations” and “occupational performance” as early as you can in your manuscript. These terms (as used in your manuscript) are probably not familiar to many readers. Most readers probably think occupational performance means workplace performance rather than the homelife reflection. The thrust of your article only became clear when I saw the statements from the OC9 scale in Appendix A. I think you need to make it known upfront that the thrust of this manuscript is understanding the impact of working conditions (as measured through the components of burnout) on a person’s daily life and well-being.
R2. We agree with the reviewer that the core terms were not properly defined. We added the definitions of “occupation” (l.60-64), “occupational performance” (l.52-54), and “daily activities” instead of “daily occupations” (l.76-78). Also, because of this, de OC9 scale name has been changed to “IDA”, in reference to “Impact on Daily Activities”.
- Throughout the article, you inconsistently use the abbreviation “DP.” Most times you present it incorrectly as “PD”.
R3. We apologize for this mistake. The full document was revised to use “DP” correctly.
- Line 38. Does “its” refer to BS? If so, simply use the term BS instead.
R4. The change was made.
- Table 2. On row 2 of the Table, I would restate it as: “Reliability if Items 7 and 8 are dropped.”
R5. We agree that the Table 2 can be understood better after this change was made, and because the selected items of the scale were not clear enough.
- Major comment. In “structural equation modeling” studies like the one performed, when the components are assembled in a manner that is statistically relevant, there needs to be some explanation (or guesses) as to why this model works. For instance, the questions I have are:
Why is it serial mediation and not parallel mediation?
Why is depersonalization the independent variable of interest and personal accomplishment and emotional exhaustion the (serial) mediators? An argument could be made that emotional exhaustion should influence depersonalization and personal accomplishment.
What I am attempting to say is your research has provided a model without any reflection on burnout (in the discussion section) - the authors do not provide any explanation as to why one would expect the components to operate in a manner that the model indicates. At least a few paragraphs should be provided trying to explain the connections. Thus, you need to attempt to explain somehow the predictive capacity of depersonalization through the mediating variables of emotional exhaustion and personal accomplishment on occupational performance. Otherwise, the results are merely just results, and perhaps a better paper would have been to simply show the correlation of a burnout index (or individual components of burnout) with occupational performance.
R6. Thanking specially for this last comment, we understand that the proposed argumentation responds to a unitary problem that is found in the manuscript. To give answer to this particular question, we have taken an interpretation, discussion, conceptualization and reflection in different parts of the study, for further clarifications. This changes will be observed in introduction (l.27-30), as well as method (l.147-148) and discussion (l.260-284). The general explanation to the proposed questions is approached in discussion. We added the following:
“The proposed serial mediation model also allows to replicate the stablished relations between the different dimensions of burnout. Starting with the analysis of Golembiewski et al. [4], the presented path analysis in our study also considers depersonalization as the trigger of the rest of dimensions of the construct. The model of Golembiewski et al. [4] stablished that a high DP also implied a high EE and low PA. In the displayed results, this relation is maintained: path A1 of the analysis points a significant relationship (coefficient -0.41), i.e., the increase of the variable DP supposes a decline of the PA. A2 effect, also significant, shows a direct relationship between DP and EE (0.98). However, even it is a classic model, its logic is being discussed in academic literature. The proposed sequence for dependence was empirically proved in teachers, but no necessarily in other analyzed work contexts [47]. As a contribution to this discussion, our path analysis supports the logic on the model from Golembiewski et al. in general population.
Another hint that encourages to deepen into this model is the fact that our design has found that the indirect effect of PA (M1) does not result significant on the displayed results. However, the indirect effect when both PA and EE are included (M1-M2), it is significant. However, it seems remarkable that the direct effect of DP on daily activities (cp) it is not statistically significant. All of this has, at least, two implications: first, a parallel mediation model between the two variables determined as mediators (EE and PA) does not result statistically relevant; and second, the direct effect (cp) is not significative, but it is the total effect. We have the certainty that in-between DP and daily activities, there are other variables intervening. In fact, the different between the direct and indirect effects showed that a 75.9% of the variance determined by DP in daily activities depends on the mediating variables. All of this helps to empower the idea of the DP as a trigger variable of the burnout [4,48], and not so much to understand that variables configuring dimensionalities of the burnout have to be observed as independent [47].”
Round 2
Reviewer 2 Report
The content of the edits improved the understanding of the context of the mediated model presented and addressed the major issues I previously had with the manuscript. However, the edits themselves were not well phrased and in some cases did not make immediate sense. So, I provided at edit of those sections as to what I thought you were attempting to convey. Please read carefully to ensure I did not change the meaning of those sections. Additional minor edits were made to the manuscript.

Author Response
We would like to thank again the reviewer the effort in revising our manuscript and also have the courtesy of reviewing our grammar and expression. We think that we added all the suggestions and the quality of the manuscript have improved, so thank again for the extensive work.